# High Olfactory Receptor-Rich 11q11 Copy Number in Girls and African American Children

**DOI:** 10.3390/genes12121943

**Published:** 2021-11-30

**Authors:** Megan Phillips, Vaithinathan Selvaraju, Anna Fouty, Jeganathan Ramesh Babu, Maninder Sandey, Thangiah Geetha

**Affiliations:** 1Department of Nutrition, Dietetics and Hospitality Management, Auburn University, Auburn, AL 36849, USA; phillml@auburn.edu (M.P.); vzs0041@auburn.edu (V.S.); adf0023@auburn.edu (A.F.); jeganrb@auburn.edu (J.R.B.); 2Boshell Metabolic Diseases and Diabetes Program, Auburn University, Auburn, AL 36849, USA; 3Department of Pathobiology, College of Veterinary Medicine, Auburn University, Auburn, AL 36849, USA; mzs0011@auburn.edu

**Keywords:** copy number, 11q11, olfactory receptor, obesity, race, gender

## Abstract

Copy number variants (CNVs) provide numerous genetic differences between individuals, and they have been linked with multiple human diseases. Obesity is one of the highly heritable complex disorders, which is associated with copy number variance (CNV). A recent report shows that the 11q11 gene, a novel olfactory receptor, and its copy number variants are involved in the early onset of obesity. In the current study, we analyzed the 11q11 gene copy number variance (CNV) based on gender in White/European American (EA) and African American (AA) normal weight and overweight/obese children. Sixty-nine boys and fifty-eight girls between the ages of 6 and 10 years belonging to either EA or AA ethnicity were involved in this study. As per World Health Organization (WHO) guidelines, each participant’s body weight and height were recorded. DNA was extracted from saliva, and the copy number variants for the 11q11 gene were measured using digital PCR. The descriptive analysis of the 11q11 copy number showed significantly more copies in girls compared to boys; similarly, AA participants had significantly increased CNV compared to EA. The normal weight (NW) and overweight/obese (OW/OB) girls were significantly less likely to belong to the low copy number variant (LCNV) group of 11q11 compared to boys; similarly, NW and OW/OB AA children were significantly less likely to belong to the LCNV group. The AA girls in LCNV had significantly higher BMI *z*-scores. Our findings suggest that the 11q11 copy number in children is race and gender-specific.

## 1. Introduction

In recent years, obesity has become an epidemic in the United States, and globally it was three times as prevalent in the year 2016 compared to the 1970s [1]. There are 340 million children and adolescents aged between 5 and 19 years who are overweight or obese, and 41 million under 5 years of age are overweight or obese, worldwide [1]. As per the Centers for Disease Control and Prevention (CDC), the obesity prevalence in the United States was 42.4% in adults during the 2017 to 2018 period among adults, and 19.3% (which is about 14.4 million) in children and adolescents [2,3]. Obesity is a major risk factor for many primary causes of mortality, including type 2 diabetes, heart disease, specific categories of cancers, and coronavirus disease 2019 (COVID-19) [4]. The development of obesity is caused by several factors, such as dietary, metabolic, environmental, and genetic. Such factors modify the gene expression related to obesity formation in children, adolescents, and adults [5]. Numerous data have revealed that body mass index (BMI) is strongly affected by genetic factors, with heritability [6]. The genetic contribution to obesity has been established through family, twin, and adoption studies. High heritability was shown in a twin study—between 40 and 77% [7]. Previous studies found single gene mutations causing obesity by involving different obesity-related pathways. Copy number variants (CNVs) are among of the under-studied genetic variations which can explain the heritability of obesity.

Copy number variance (CNV) pertains to a segment of DNA one kilobase (kb) long or more, which can have different copy numbers in individuals of the same species [8]. CNV of DNA sequences from 270 individuals with four different populations was first identified in 1447 CNV regions [9]. A recent update from the Sangers Institute database shows more than 38,092 CNV from 36,000 patients [10]. Though CNVs are not directly caused by disease, there are several occurrences CNVs affecting critical developmental genes that cause disease. The distal CNVs of 16p11.2 determine the basal ganglia volume in developing neurodevelopmental disorders [11,12]. An investigation of the deletion or duplication of 16p11.2 in eating behavior, cognitive traits, and BMI suggested that the satiety response in children is related to BMI. This was also observed in adults [13].

Further analysis involved mutation screening and CNV analysis of 10q11.22 among 326 children and adolescents with obesity and 298 normal individuals. The results revealed a lower copy number in the obese population [14]. The combined analysis of single nucleotide polymorphism (SNP) and CNV conducted on European Americans (EA) and African Americans (AA) showed an association of BMI with only 16p12.3 CNV among the 84 previously identified CNV regions [15]. Genomic CNVs using Affymetrix genome-wide human SNP array 6.0 showed a similar association between BMI and 16p12.3 in European and Chinese populations [16]. Previous studies on obesity-related CNVs and dietary behaviors among control and obese children showed that unusual copy numbers suggested genetic susceptibility to obesity [17]. Our previous study demonstrated that overweight/obese children expressed low salivary *AMY1* CNV, and copy number inversely correlated with obesity, especially in AA children [18].

The rapid development of obesity occurs due to the overeating of high-energy foods when they are not required for regular physiological needs [19]. It happens due to external stimuli, especially the smell of foods that elicit appetite and increase food consumption [20]. The olfactory system plays a significant role in unintentional food consumption, which is a major contributor to obesity. Previous studies showed that high BMI is correlated with olfactory dysfunction [21,22]. The 11q11 locus is approximately 80 kb and covers three functional olfactory receptor (OLR) genes, all belonging to OLR family 4: *OR4P4*, *OR4S2*, and *OR4C6* [23]. Few studies have shown the relationship of the 11q11 locus with obesity [17,24,25,26]. This study aimed to evaluate the 11q11 copy number variation by gender, race (EA and AA), and overweight/obese status, in children. 

## 2. Materials and Methods

### 2.1. Study Participants

The cross-sectional study consisted of 69 boys and 58 girls between the ages of 6 and 10 years belonging to either EA or AA ethnicity. Power analysis was done using G Power 3.1, and the *t*-test was a two-tailed statistical test with a significance level of 5%. The total number of children recruited into the study was 127. With the sample sizes, the assumption of a two-group *t*-test, and the prior assumptions, we had 80% power to detect effect sizes of 0.5 (d)—and greater for comparing pairs of the racial groups’ weight-based groups—as statistically significant. An initial survey by phone or email was conducted with the parents of interested participants to discover the existence of major health problems, such as diabetes, cardiovascular disease, or taking long-term medications, which were disqualifying for the study. The children were brought to Auburn University to participate, and written consent was collected from the parents and participants. Auburn University Institutional Review Board approval was obtained.

### 2.2. Anthropometric Measurement of the Participants

Participants’ body weights and heights were recorded as per the World Health Organization (WHO) guidelines. A child was categorized as underweight if his/her BMI percentile was <5, normal weight if the percentile was between ≥5 to ≤85, overweight between >85 and ≤95, and obese if the percentile was >95 according to the standardized growth curves developed by CDC [27]. 

### 2.3. 11q11 Copy Number Measurement by 3D Digital PCR

The participants provided saliva samples via the DNA GenoTek saliva collection kit (Ottawa, ON, Canada). Isolation of genomic DNA from salivary samples was performed with the PrepIT-L2P method (DNA GenoTek, Ottawa, ON, Canada). Briefly, the saliva in the DNA GenoTek kit was mixed well and incubated in the water bath at 50 °C for a minimum of 1 h to release the DNA, and then the nucleases were inactivated. PrepIT-L2P (20 µL) was added to the saliva sample, incubated in ice for 10 min, and centrifuged (15,000× *g*) for 5 min. The supernatant containing the DNA was separated and mixed with absolute alcohol (600 µL), and the DNA precipitate formed was left to sit at room temperature (RT) for 10 min. The samples were centrifuged for 2 min, and the pellet containing DNA was separated. The pellet was washed with 70% ethanol and centrifuged (15,000× *g*) at RT for 1 min; then we thoroughly removed the ethanol. TE buffer (75 µL) was added to the pellet, the DNA sample was left at RT overnight for complete rehydration and used for 3D digital PCR analysis. 

The copy number of the 11q11 gene was analyzed using digital PCR (Quantstudio 3D Digital PCR). This reaction mix was prepared with two TaqMan assays, 11q11 (Hs03802074_cn, FAM-labeled; amplicon length 109) and a reference gene (RNase P, VIC labeled), along with the 3D digital PCR master mix and diluted DNA (10 ng/µL). From the reaction mix, 14.5 µL was loaded into the QuantStudio 3D Digital PCR chip with the help of a chip loader. Chips were loaded into the ProFlex 2X Flat PCR system with the PCR cycling conditions of initial denaturation at 96 °C for 10 min, followed by 39 cycles at 60 °C for 2 min and 98 °C for 30 s; one cycle at 60 °C incubation for 2 min; and infinite holding at 4 °C. Once PCR was completed, the chips were scanned using the QuantStudio 3D Digital PCR System and data analyzed using QuantStudio 3D AnalysisSuite software (https://www.thermofisher.com/us/en/home/life-science/pcr/digital-pcr/quantstudio-3d-digital-pcr-system/quantstudio-3d-software.html; accessed on 20 March 2020). The copy numbers of 11q11 in the samples were calculated using the reference gene RNAase P. 

### 2.4. Statistical Analysis

An assessment of the normality of the data was done by Kolmogorov–Smirnov test, and the data skewness, kurtosis, showed normal distributions for the overall data of the participants, and the data of age, height, and CNV. The data were grouped by CNV into LCNV (0 or 1 CNV) and HCNV groups (2 to 4 CNVs) based on the number of copies present. The CNV grouped data showed normal distributions of age and height, and the other variables, including CNV, were non-normally distributed. The normally distributed sample’s skewness falls within the liberal z-range of samples with 50–300 participants (±3.29) [28,29]. We analyzed a two-way ANOVA for age, height, BMI, weight, and CNV using CNV groups with other covariates—gender, race, and BMI category—as the grouping variables. The cross-tabulation and binary logistic regression analysis were performed with the respective covariates to analyze the proportional graphs data. The bar graphs were prepared with GraphPad Prism V 8.2.0, and a student *t*-test was used to measure the BMI z-score difference between the two groups. The statistical analysis was performed with SPSS 25 software (IBM, Armonk, NY, USA).

## 3. Results

The study consisted of 127 participants; 68 EA and 59 AA provided their salivary samples for 11q11 gene analysis. The same participants were used in our previous study, and the anthropometric measurements showed significant differences in the overweight/obese (OW/OB) group compared to the normal weight (NW) group in weight, BMI, BMI z-score, and waist circumference [18]. The participants were separated into 11q11 low copy number and high copy number. The ages, BMI, weights, heights, and 11q11 copy numbers of participants with low and high copy number variants were analyzed by two-way ANOVA, and the results are provided in Table 1. The differences between LCNV and HCNV groups and their interaction effects with variates are reported in the table with F-statistics and *p*-values for age, height, weight, BMI, and CNV. 

The 11q11 CNV descriptive analysis data (Figure 1A) showed a significantly higher (*p* = 0.025) mean 11q11 copy number in girls (1.724 ± 0.09) compared to boys (1.473 ± 0.07). The 11q11 copy number ranged from zero to four. The distributions of copy number rounded to the nearest integer in boys and girls are provided in Figure 1B. We analyzed the percentages NW and OW/OB boys and girls. Even though the fewer girls were NW (43.42%) than boys (56.58%), the difference was not statistically significant (Figure 1C). However, we found significantly more girls in HCNV than LCNV (*p* = 0.031). As shown in Figure 1D, boys did not have significantly different proportions of HCNV and LCNV. However, girls were significantly more likely to belong to HCNV (64.71%) than LCNV (35.29%). Figure 1E shows that the NW (38.24%) and OW/OB (33.33%) girls had significantly lower proportions of participants with a LCNV compared to boys. However, in the HCNV group, the OW/OB girls (57.58%) proportion was not significantly higher than that of the OW/OB boys (42.42%). 

Similarly, we analyzed the difference in the mean 11q11 CNV between white/European American (EA) and African American (AA) children. Figure 2A shows that AA participants (1.716 ± 0.09) had a significantly higher (*p* = 0.031) 11q11 copy number compared to EA participants (1.476 ± 0.07). The race-specific frequency distribution of CNV is shown in Figure 2B. Two of the AA participants had four 11q11 copies, whereas no EA did. We analyzed the proportions of participants in NW and OW/OB groups in EA and AA participants and found no significant difference, as shown in Figure 2C. There was significantly higher (*p* < 0.0001) proportion of HCNV in AA children than LCNV (Figure 2D). As shown in Figure 2E, a significant difference was observed between the proportions of NW and OW/OB groups of EA and AA participants with LCNV. The results suggest that AA participants had a significantly higher mean 11q11 copy number compared to the EA participants, and an increased HCNV proportion, but there was no significant difference in CNV between NW and OW/OB children of AA.

Figure 3A shows the mean 11q11 CNV among EA and AA in boys and girls. There was no significant difference observed in EA and AA among the boys and girls (Figure 3A). The race-specific frequency distributions of 11q11 copy numbers in girls and boys are shown in Figure 3B. More AA girls had two and four 11q11 copies compared to EA participants. Figure 3C shows the proportional differences between LCNV and HCNV among NW and OW/OB groups, race, and gender. In EA, the percentage of HCNV was significantly higher (*p* = 0.019) in OW/OB girls (33.33%) than the percentage of LCNV girls overall (7.41%). In AA, there was a significant difference in the LCNV and HCNV proportions between OW/OB boys and girls. NW AA girls were significantly differently distributed in LCNV and HCNV groups (*p* = 0.013). In boys, the proportion of LCNV in OW/OB AA participants was significantly lower than in OW/OB EA participants (*p* = 0.017). A comparison of proportions between the LCNV and HCNV showed significantly more HCNV prevalence (*p* < 0.035) in OW/OB AA boys (33.33%) compared to low copy number variant prevalence (8.33%). AA girls in NW (11.43% vs. 37.14%) and OW/OB groups (16.67% vs. 41.67%) were significantly more likely to fit into HCNV than LCNV. The results suggest that the higher 11q11 copy numbers in girls are mainly due to the higher proportions of OW/OB of EA and AA, and NW of EA, being in the HCNV group.

BMI *z*-scores of boys and girls of AA and EA ethnicities with LCNV and HCNV were analyzed with a *t*-test and are expressed as bar graphs in Figure 4. LCNV and HCNV did not show a significant difference in BMI *z*-score between EA and AA boys and girls (Figure 4).

## 4. Discussion

We have investigated the 11q11 copy numbers in the salivary DNA of girls and boys aged 6–10 years old of EA and AA ethnicities. Table 1 discusses the differences in the age, BMI, and 11q11 copy number of participants with low and high copy number variants of 11q11. The mean copy number of 11q11 was found to be higher in girls compared to boys. The proportion of participants with high CNV among girls correlates with a study conducted on 150 individuals of three different populations that showed gender specificity in the CNV of the olfactory receptor gene family [30]. A previous study conducted in children of German descent suggested a genetic link between obesity and 11q11 copy numbers [24]. A family-based genome-wide study revealed a lower 11q11 copy number in obese children than normal weight ones [24]. Our results showed a higher BMI z-score in AA girls with LCNV compared to boys with LCNV (Figure 4A), which correlates with previously published data. Increased BMI was also found to be associated with a low 11q11 copy number in Chinese children [17]. Zhang et al. observed a significant increase in the risk of obesity in Chinese children with deletions at the 11q11 locus. In addition, a cumulative effect was seen in Chinese children when deletions accompanied this at two other loci related to obesity: 10q11.22 and 4q25. This indicates a robust cumulative association between loci with high copy numbers and childhood obesity [17]. In contrast, León-Mimila et al. reported a positive association in Mexican children aged 6–12 years. A lower 11q11 copy number was significantly associated with lower obesity risk in children. The same association was not seen in adult subjects [25]. We separated the children into 11q11 low and high copy number groups. There was no significant difference in 11q11 copy number (according to LCNV and HCNV groups) between NW and OW/OB children. Interestingly, when the participants were separated based upon gender, the NW and OW/OB boys were significantly more likely to belong to the LCNV group than the NW and OW/OB girls. This result suggests that the difference in the 11q11 copy number is gender-specific in children.

Significant discrepancies exist between people of different ethnicities within a specific population. Hence, we also assessed the 11q11 copy number by race, and found that the AA participants had significantly higher copy numbers than EA children. Shadravan (2013) in his study conducted on three different races (East Asian, EA, and AA), observed that the gender difference in 11q11 CNV is extensively dependent on ethnicity, and it was absent in the East Asian population [30]. The genome wide CNV survey conducted on people with European ancestry showed large copy numbers in obese patients [31]. D’Angelo and Koiffmann’s (2012) review on GWAS CNV in individual participants with high and rare CNVs showed a significant risk of obesity, especially when deleting 16p112 [32]. Ethnic differentiation of the CNV of 16p12.3 reported for obese phenotypes of European and Chinese populations may support our findings regarding race [16].

The GWAS analysis showed the importance of CNV to childhood obesity participants of European and African ancestry [33]. The results show that in EA, the OW/OB girls are significantly less likely to belong to the 11q11 LCNV group than the OW/OB boys. Additionally, the OW/OB EA girls are significantly more likely to have a HCNV than a LCNV. AA participants were more likely to have a HCNV, except NW boys. AA girls had higher BMI *z*-score compared to boys with LCNV. This suggests that the 11q11 copy number in children is race and gender-specific.

There are a few limitations to this study worth noting. The results were obtained from a small number of participants, and their diets were not included in the study. A larger number of participants would be ideal for a more indicative sample. 

## 5. Conclusions

In summary, our findings regarding salivary 11q11 copy number frequency showed significantly higher copy numbers in girls and AA children. The LCNV of 11q11 was significantly less common in NW and OW/OB girls and AA. The AA girls with LCNV had significantly higher BMI z-scores. The 11q11 copy number in children is race and gender-specific. 

## Figures and Tables

**Figure 1 genes-12-01943-f001:**
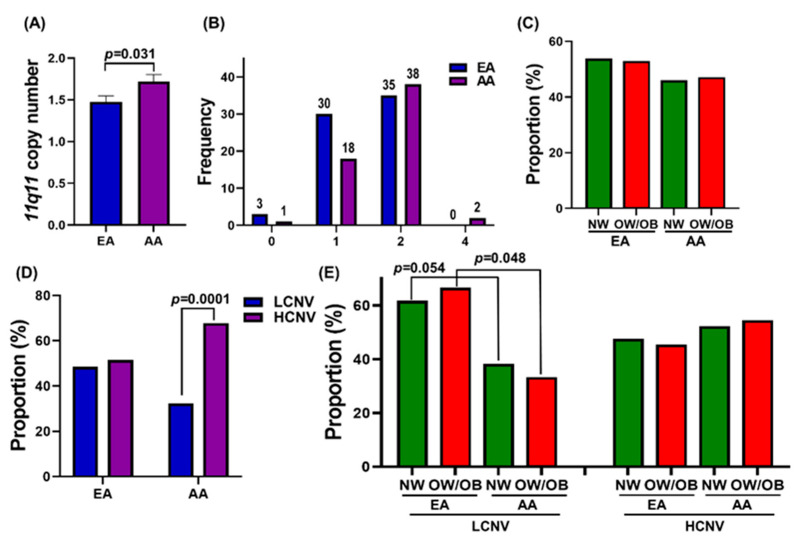
11q11 copy number in boys and girls. (**A**) The bar graphs represent the mean ± SEM of 11q11 copy number in boy and girl participants. (**B**) Distribution of 11q11 copy number in boys and girls. (**C**) The proportions boys and girls with NW and OW/OB. (**D**) The proportions of low and high CNV in boys and girls. (**E**) The differences in proportions among NW and OW/OB boys and girls, with LCNV and HCNV.

**Figure 2 genes-12-01943-f002:**
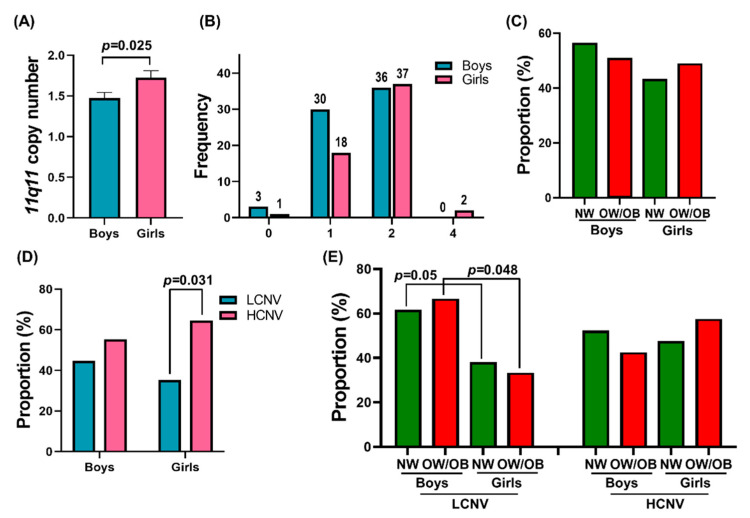
11q11 copy number in White/European American (EA) and African American (AA) children. (**A**) The bar graphs represent the mean 11q11 copy number differences between EA and AA participants. (**B**) Distribution of 11q11 copy number in the study population by race. (**C**) The proportions of participants with NW and OW/OB in EA and AA. (**D**) The proportions of low and high CNV among EA and AA. (**E**) The proportions of NW and OW/OB in EA and AA, with LCNV and HCNV.

**Figure 3 genes-12-01943-f003:**
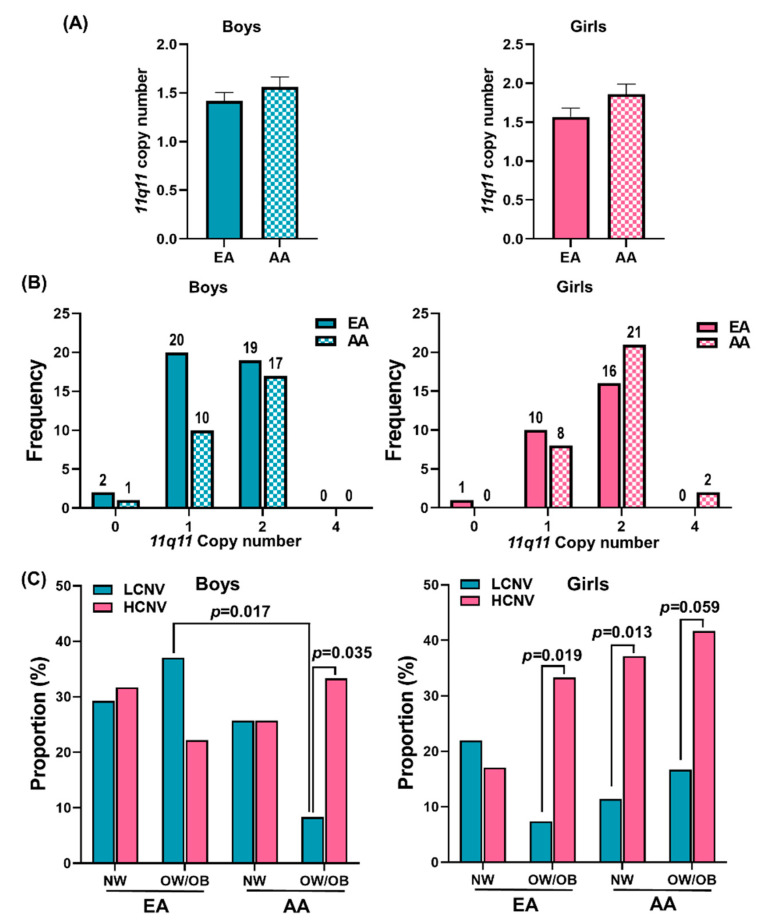
11q11 copy numbers in boys and girls of White/European American (EA) and African American (AA) ethnicities. (**A**) The bar graphs represent the mean 11q11 copy number differences between gender-specific EA and AA participants. (**B**) Distribution of 11q11 copy number in the EA and AA boys and girls. (**C**) The differences among NW and OW/OB EA and AA participants of both genders in terms of LCNV and HCNV.

**Figure 4 genes-12-01943-f004:**
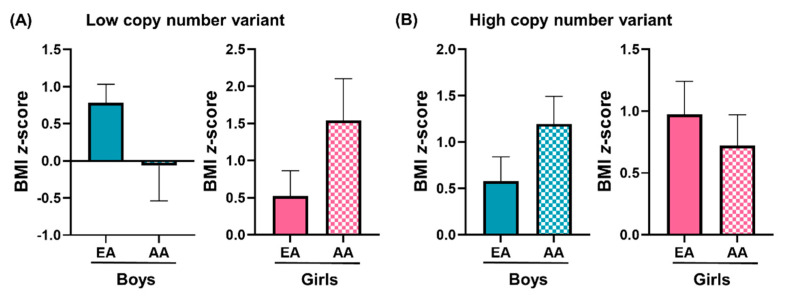
BMI *z*-scores of boys and girls with low copy number variant (LCNV) and high copy number variant (HCNV) of White/European American (EA) and African American (AA) ethnicities. (**A**). BMI *z*-scores of EA and AA boys and girls with LCNV. (**B**) BMI *z*-scores of EA and AA boys and girls with HCNV.

**Table 1 genes-12-01943-t001:** General characteristics and 11q11 copy number in the study population.

		11q11 Low Copy Number Variant (LCNV, *n* = 52)	11q11 High Copy Number Variant (HCNV, *n* = 75)	F (1, 127)	*p*-Value
**Age (year)**	All Participant	8.22 ± 0.19	8.53 ± 0.16	1.558	*p* = 0.214
Race (*n*)	EA (68)	8.26 ± 0.24	8.39 ± 0.24	0.113	*p* = 0.738
	AA (59)	8.16 ± 0.32	8.66 ± 0.22		
		F (1, 127) = 0.541; *p* = 0.464			
Gender (*n*)	Boys (69)	8.14 ± 0.24	8.45 ± 0.23	0.508	*p* = 0.477
	Girls (58)	8.36 ± 0.32	8.60 ± 0.22		
		F (1, 127) = 0.016; *p* = 0.898			
BMI category (*n*)	NW (76)	8.10 ± 0.24	8.55 ± 0.21	0.372	*p* = 0.543
	OW/OB (51)	8.45 ± 0.33	8.51 ± 0.24		
		F (1, 127) = 0.566; *p* = 0.453			
**Height (cm)**	All Participant	130.27 ± 1.56	132.81 ± 1.30	1.564	*p* = 0.213
Race	EA	129.40 ± 1.95	130.22 ± 1.89	3.057	*p* = 0.083
	AA	131.77 ± 2.57	135.08 ± 1.77		
		F (1, 127) = 0.365; *p* = 0.547			
Gender	Boys	129.02 ± 1.96	131.49 ± 1.88	2.043	*p* = 0.155
	Girls	132.43 ± 2.59	134.03 ± 1.81		
		F (1, 127) = 0.043; *p* = 0.836			
BMI category	NW	127.78 ± 1.88	130.46 ± 1.69	9.369	*p* = 0.003
	OW/OB	134.95 ± 2.58	135.79 ± 1.91		
		F (1, 127) = 0.203; *p* = 0.653			
**Weight (kg)**	All Participant	32.15 ± 1.53	32.29 ± 1.27	0.004	*p* = 0.947
Race	EA	29.79 ± 1.89	30.46 ± 1.83	6.118	*p* = 0.015
	AA	35.26 ± 2.48	33.88 ± 1.71		
		F (1, 127) = 0.578; *p* = 0.448			
Gender	Boys	32.72 ± 1.93	31.86 ± 1.85	0.033	*p* = 0.856
	Girls	31.16 ± 2.54	32.68 ± 1.78		
		F (1, 127) = 0.340; *p* = 0.561			
BMI category	NW	31.33 ± 1.90	32.45 ± 1.71	0.233	*p* = 0.630
	OW/OB	33.71 ± 2.61	32.07 ± 1.93		
		F (1, 127) = 0.447; *p* = 0.505			
**BMI (kg/m^2^)**	All Participant	17.89 ± 0.50	18.28 ± 0.41	0.363	*p* = 0.548
Race	EA	17.50 ± 0.62	17.72 ± 0.60	2.667	*p* = 0.105
	AA	18.58 ± 0.82	18.78 ± 0.56		
		F (1, 127) = 0.0004; *p* = 0.984			
Gender	Boys	17.25 ± 0.62	18.27 ± 0.59	1.870	*p* = 0.174
	Girls	19.02 ± 0.82	18.29 ± 0.57		
		F (1, 127) = 1.787; *p* = 0.184			
BMI category	NW	15.74 ± 0.40	16.10 ± 0.36	161.89	*p* < 0.001
	OW/OB	21.97 ± 0.56	21.06 ± 0.41		
		F (1, 127) = 2.098; *p* = 0.150			
**11q11 CNV**	All Participant	0.928 ± 0.04	2.05 ± 0.04	407.86	*p* < 0.001
Race	EA	0.917 ± 0.05	2.00 ± 0.05	0.928	*p* = 0.337
	AA	0.948 ± 0.07	2.08 ± 0.05		
		F (1, 127) = 0.163; *p* = 0.687			
Gender	Boys	0.916 ± 0.05	1.98 ± 0.05	1.777	*p* = 0.185
	Girls	0.950 ± 0.07	2.10 ± 0.05		
		F (1, 127) = 0.527; *p* = 0.469			
BMI category	NW	0.918 ± 0.05	2.09 ± 0.05	0.284	*p* = 0.595
	OW/OB	0.948 ± 0.07	1.99 ± 0.05		
		F (1, 127) = 1.113; *p* = 0.294			

European American (EA); African American (AA); normal weight (NW); overweight/obese (OW/OB); copy number variants (CNV); values are expressed as mean ± SEM; two-way ANOVA was used to calculate the data presented in the table.

## Data Availability

The data presented in this study are available on request from the corresponding author.

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
