# Peer review of "High Olfactory Receptor-Rich 11q11 Copy Number in Girls and African American Children"

_genes, 2021, doi:10.3390/genes12121943_

Round 1

Reviewer 1 Report

This study assess the copy number of the 11q11 region containing three olfactory receptors in relation to obesity parameters, race and gender in children. The hypothesis that olfactory receptor copy number is  affecting obesity is valid and supported by previous research. However, I have some concerns regarding the sample selection and the statistics.

In section 2.4 the distribution of the data is not stated, the types of t-test and one-way ANOVA (post test?). The cut off value for assigning samples into high or low copy numbers is not mentioned. There is no reference to the online calculater used. With a population with factors gender and race, a T test would not be the best choice. A two way ANOVA would be more appropriate. 

Table 1 is difficult to read. In the top section of the table the continuous variables age, BMI, weight and BMI is listed as mean+SEM of the groups LCNV and HCNV. In the bottom section the table appear to be the measured copy number in the beforementioned groups but now means are within the subcategories gender, race and BMI category. It seems as If a low copynumber would be 1 and and high is around 2. This do nook look like continuous data to be, and based on the distribution of copy numbers fig 1b and fig 2b there are virtually no copy numbers of 0 or 4 in the dataset. I would exclude the values 0 and 4 from the analysis. The outcome variable now look binary (1 or 2) and I would consider that in the statistical analysis. Also in table 1, it is not mentionen how the p values are calculated. the p values in the bottom section are all very low, but in the plots in fig 1-3 only one p value of 0.001 is shown. This make me question the whole setup and the calculations behind these p values. Please redo all the statistical analysis and provide the test behind each calculated p value.

In addition, some sentences in the manuscripts must be improved. Examples are line 52, where you miss the word "individuals" or "patients".  Sentences starting at line 132 and 134 are also difficult to comprehend due to missing details in the statistical analysis.

Author Response

Reviewer 1

This study assess the copy number of the 11q11 region containing three olfactory receptors in relation to obesity parameters, race and gender in children. The hypothesis that olfactory receptor copy number is affecting obesity is valid and supported by previous research. However, I have some concerns regarding the sample selection and the statistics.

Question: In section 2.4 the distribution of the data is not stated, the types of t-test and one-way ANOVA (post-test?). The cut off value for assigning samples into high or low copy numbers is not mentioned. There is no reference to the online calculator used. With a population with factors gender and race, a T test would not be the best choice. A two-way ANOVA would be more appropriate.

Answer: The statistical analysis section was modified as suggested by the reviewer to reflect the distribution of the CNV and other statistical methods. The gender and race (Figure 1A and 2A) were calculated by t-test, and the rest of the graphs were derived based on the percentage proportion calculation by Medcalc statistical online software. The citation for the online calculator is added to the statistical analysis under the Methods section. The t-test, one-way ANOVA, and online statistical calculations were used based on the requirement of statistical analysis. The results of populations of different gender and race are discussed in the proportional graphs. It is not the mean of a quantitative variable, and for this kind of proportional graph, Medcalc is more suitable than two-way ANOVA (Page 3, line 129-140).

Question: Table 1 is difficult to read. In the top section of the table the continuous variables age, BMI, weight and BMI is listed as mean+SEM of the groups LCNV and HCNV. In the bottom section the table appear to be the measured copy number in the beforementioned groups but now means are within the subcategories gender, race, and BMI category. It seems as If a low copynumber would be 1 and and high is around 2. This do nook look like continuous data to be, and based on the distribution of copy numbers fig 1b and fig 2b there are virtually no copy numbers of 0 or 4 in the dataset. I would exclude the values 0 and 4 from the analysis. The outcome variable now look binary (1 or 2) and I would consider that in the statistical analysis. Also in table 1, it is not mentionen how the p values are calculated. the p values in the bottom section are all very low, but in the plots in fig 1-3 only one p value of 0.001 is shown. This make me question the whole setup and the calculations behind these p values. Please redo all the statistical analysis and provide the test behind each calculated p value.

Answer: As suggested by the reviewer, the second part of the table discusses the mean+SEM of 11q11 CNV. In the 11q11 CNV analysis, deletion, and duplication of CNV are important. Thus, even though 0 and 4 CNV were less frequent in the study participants, we included them in the analysis. The range of 11q11 CNV is 0-8, and it is not a binary (Leon-Mimila et al., Nutrients 2018, 10, doi:10.3390/nu10111607). In table 1, p-values are calculated by unpaired t-test, and part 2 of the table reflects it. We have modified the table footnote accordingly. The p values of Figures 1-3 are calculated based on the percentage proportion. The online Medcalc calculator is used to calculate the p values, and its reference is included in the statistical analysis section (Page 3, line 139-140). Following are the references included in the manuscript.

  1. Campbell, I. Chi-squared and Fisher–Irwin tests of two-by-two tables with small sample recommendations. Statistics in Medicine 2007, 26, 3661-3675, doi:https://doi.org/10.1002/sim.2832.
  2. Richardson, J.T. The analysis of 2 x 2 contingency tables--yet again. Stat Med 2011, 30, 890; author reply 891-892, doi:10.1002/sim.4116.

Question: In addition, some sentences in the manuscripts must be improved. Examples are line 52, where you miss the word "individuals" or "patients". Sentences starting at line 132 and 134 are also difficult to comprehend due to missing details in the statistical analysis.

Answer: As suggested by the reviewer the sentences are modified as shown in Line No. 52, and 132-134).

Reviewer 2 Report

In this paper, the authors found that "High Olfactory Receptor-Rich 11q11 Copy Number in Girls and African American Children". However, the results from this paper did not support this conclusion. 

A major issue of this paper is unbalanced sample size. There are 41 boys and 27 girls for EA and 28 boys and 31 girls for AA. There are more boys than girls for EA. Any bias during data collection?

Assuming one of the authors' conclusion is true that girls have high copy number. The girl proportion is high in AA than EA. High copy number in AA (Figure 2A) can be explained by more girls in AA instead of high copy number in AA. A simple example, all girls have copy number of 2; all boys have copy number of 1; no difference between AA and EA. If there are more boys in EA, then we will see high copy number is AA. Similar for Figure 2D, there is no difference between LCNV and HCNV in EA might because there are more boys in EA. If the sample size of boys and girls is similar in EA, we might see the difference. Both significant and unsignificant results could be caused by the bad experiment design of this study. The results are unreliable. Similarly, if that AA has high copy number is true, we could get high copy number in girls because there are more girls in AA. From the results presented in the paper, we CANNOT conclude "High Olfactory Receptor-Rich 11q11 Copy Number in Girls and African American Children".

If the authors want to conclude there are more copy number in girl, they have to compare boys to girls in EA and AA separately. They also have to separate boys and girls when comparing EA and AA. Also, please double check whether there are any bias during data collection. 

Author Response

Reviewer 2

In this paper, the authors found that "High Olfactory Receptor-Rich 11q11 Copy Number in Girls and African American Children". However, the results from this paper did not support this conclusion. 

Question: A major issue of this paper is unbalanced sample size. There are 41 boys and 27 girls for EA and 28 boys and 31 girls for AA. There are more boys than girls for EA. Any bias during data collection? Assuming one of the authors' conclusion is true that girls have high copy number. The girl proportion is high in AA than EA. High copy number in AA (Figure 2A) can be explained by more girls in AA instead of high copy number in AA. A simple example, all girls have copy number of 2; all boys have copy number of 1; no difference between AA and EA. If there are more boys in EA, then we will see high copy number is AA. Similar for Figure 2D, there is no difference between LCNV and HCNV in EA might because there are more boys in EA. If the sample size of boys and girls is similar in EA, we might see the difference. Both significant and unsignificant results could be caused by the bad experiment design of this study. The results are unreliable. Similarly, if that AA has high copy number is true, we could get high copy number in girls because there are more girls in AA. From the results presented in the paper, we CANNOT conclude "High Olfactory Receptor-Rich 11q11 Copy Number in Girls and African American Children".

Answer: There is no bias in recruiting the participants. Actual participant numbers are presented in the study, we had more EA boys than girls participating in this study.

Even though the number of girls’ and boys’ participants are not the same, Figure 1A shows that overall girls have a greater mean of 11q11 copy number compared to boys. Figure 2A shows that AA has a greater mean 11q11 copy number compared to EA. Hence, we are suggesting that there is a high 11q11 copy number in overall girls and African American children.

Question: If the authors want to conclude there are more copy number in girl, they have to compare boys to girls in EA and AA separately. They also have to separate boys and girls when comparing EA and AA. Also, please double check whether there are any bias during data collection. 

Answer:  According to the reviewer’s suggestion we have separated boys and girls in EA and AA separately in Figure 3. We found no significant difference in the mean of 11q11 copy number between boys and girls of both EA and AA population (Figure 3A). Hence, we are not concluding that there is a high 11q11 copy number in girls belonging to AA ethnicity.

Round 2

Reviewer 1 Report

I asked about the distribution of the data and that was not provided. To specify, I was interested in whether the data is normally distributed (age, bmi weight) which may be skewed by the covariates like sex (/gender). These variables may be normally distributed, but I would like that to have been mentioned. The Low versus High copy number variants is definitely not normally distributed as the low variant group contains mainly 1's and the high variant group contain mainly 2's. Fig 1B shows that the outcome is nearly binary (if the outliers with 0 and 4 are removed) or categorical at best (values 0,1,2,4). Applying a T test to this type of data as you state in table 1 is simply wrong. Also, I fail to see where the mentioned one-way anova is applied.

I suggested using a 2-way ANOVA for the age, height and BMI outcome, and use CNV group and the other covariates like gender and race as the grouping variables. For analysing the CMV group as an outcome as you do in fig 1-3 you need to apply a logistic regression model or equivalent if you wish to keep the 4 CNV categories and make a model using all your covariates. You mention a medcalc online calculator, and for the newly provided references I see that is is a chi square test. This could just as well be calculated in your statistics program of choice and the actual test should have been mentioned in the methods section, as the used test is the information essential for understanding how the p values was calculated, not the calculator itself. I reject the paper based on the poorly described statistics and the t test applied to non-normally distributed data. Also, arguing that a CNV count of 0-8 seen in another population makes your variable non-binary is maybe right for the dataset you refer to, but it is still a categorial response. It can behave like a continuous response if your range is wide enough and your count are evenly distributed but in your dataset the majority of the values are 1 and 2 and then you have a few counts that are 0 and 4. 

The grouping sentence added to the statistical analysis section (line 137) is confusing as it does not include the nr 2 in the groups, but it is clearly present in the HCNV based on the numbers in table on and on the plots.

Author Response

Reviewer 1

Question: I asked about the distribution of the data and that was not provided. To specify, I was interested in whether the data is normally distributed (age, bmi weight) which may be skewed by the covariates like sex (/gender). These variables may be normally distributed, but I would like that to have been mentioned. The Low versus High copy number variants is definitely not normally distributed as the low variant group contains mainly 1's and the high variant group contain mainly 2's.

Answer: We agree with the reviewer. The overall participants' data of age, height and CNV are normally distributed, and their skewness falls under the liberal z range of sample size between 50 and 300 participants. When data is separated into low and high copy number variants, only age and height are normally distributed, and the rest of the data, including CNV, are non-normally distributed. As suggested by the reviewer, we have included this in the statistical section (Page No. 3; Line No. 130-132).

Question: Fig 1B shows that the outcome is nearly binary (if the outliers with 0 and 4 are removed) or categorical at best (values 0,1,2,4). Applying a T test to this type of data as you state in table 1 is simply wrong. Also, I fail to see where the mentioned one-way ANOVA is applied. I suggested using a 2-way ANOVA for the age, height and BMI outcome, and use CNV group and the other covariates like gender and race as the grouping variables.

Answer: As suggested by the reviewer, we considered the variables as categorical and analyzed them. Table 1 was modified by analyzing a two-way ANOVA for age, height, weight, BMI, and CNV using low and high CNV groups with other covariates, gender, race, and BMI category as the grouping variables.

Question: For analysing the CMV group as an outcome as you do in fig 1-3 you need to apply a logistic regression model or equivalent if you wish to keep the 4 CNV categories and make a model using all your covariates. You mention a medcalc online calculator, and for the newly provided references I see that is a chi square test. This could just as well be calculated in your statistics program of choice and the actual test should have been mentioned in the methods section, as the used test is the information essential for understanding how the p values was calculated, not the calculator itself. I reject the paper based on the poorly described statistics and the t test applied to non-normally distributed data. Also, arguing that a CNV count of 0-8 seen in another population makes your variable non-binary is maybe right for the dataset you refer to, but it is still a categorial response. It can behave like a continuous response if your range is wide enough and your count are evenly distributed but in your dataset the majority of the values are 1 and 2 and then you have a few counts that are 0 and 4. 

Answer: We agree with the reviewer, the categorical variables used cross-tabulation and binary logistic regression analysis for the proportional graph significance calculation. All the analysis was done using SPSS instead of medcalc online calculator. There was no change in the proportional graphs. The statistical analysis section was updated in the revised version (Page No. 3; Line No. 138-140).

Question: The grouping sentence added to the statistical analysis section (line 137) is confusing as it does not include the nr 2 in the groups, but it is clearly present in the HCNV based on the numbers in table on and on the plots.

Answer: As the reviewer pointed, the grouping sentence was modified in the statistical analysis section. The sentence added on Page No. 3; Line No. 132-133 is as follows “The CNV is separated as LCNV (0 and 1 CNV) and HCNV group (2 and 4 CNV) based on the number of copies present”.

Reviewer 2 Report

A simple example, comparing copy number changes on chromosome X. Girls have two copies and boys have one copy. Girls have more copies than boys on chromosome X. There is a study investigating copy number difference between EA and AA on chromosome X. They recruit 100 EA girls and 100 AA boys and then they conclude that EA has more copies than AA on chromosome X. 

Although this is an extreme example, there is similar issue in this paper. We can pool boys and girls together when comparing EA and AA ONLY when we assume there is no difference between boys and girls. We can pool EA and AA together when comparing boys and girls ONLY when we assume there is no difference between EA and AA. However, according to the title of this paper, copy number is different between girls and boys and different between EA and AA. Then we cannot pool them to take the tests. The authors have to stratify the analysis. When comparing EA to AA, they have to comparing EA boys to AA boys and then EA girls to AA girls. Similar procedure has to be taken when comparing boys to girls. 

Author Response

Reviewer 2

A simple example, comparing copy number changes on chromosome X. Girls have two copies and boys have one copy. Girls have more copies than boys on chromosome X. There is a study investigating copy number difference between EA and AA on chromosome X. They recruit 100 EA girls and 100 AA boys and then they conclude that EA has more copies than AA on chromosome X. 

Question: Although this is an extreme example, there is similar issue in this paper. We can pool boys and girls together when comparing EA and AA ONLY when we assume there is no difference between boys and girls. We can pool EA and AA together when comparing boys and girls ONLY when we assume there is no difference between EA and AA. However, according to the title of this paper, copy number is different between girls and boys and different between EA and AA. Then we cannot pool them to take the tests. The authors have to stratify the analysis. When comparing EA to AA, they have to comparing EA boys to AA boys and then EA girls to AA girls. Similar procedure has to be taken when comparing boys to girls. 

Answer: Thank you for the reviewer's suggestion. In Figure 1, all the boys are compared to girls without separating them into the race. Similarly, in Figure 2, the overall EA was compared to AA participants without separating into boys and girls. We modified Figures 3 and 4 and compared the EA to AA boys and EA to AA girls and updated the figure and results.

This manuscript is a resubmission of an earlier submission. The following is a list of the peer review reports and author responses from that submission.

Round 1

Reviewer 1 Report

In this paper, Phillips et al. analyzed the 11q11 copy number in 127 participants, including 68 EA and 59 AA. They found that the 11q11 copy number in the overweight/obese children is race and gender specific.

Overall, I think the study design, data analysis, and writing were significantly flawed and the data were not convincing at all. First, I don’t understand why they only focused their analysis on 11q11 copy number. Have they performed any genome-wide copy number variant analysis to systematically evaluate the contribution of this type of variant to the phenotype of interest? The pricing of genotyping array is very low for this type of study nowadays. Second, I felt that authors just tried to perform statistical comparisons their current data set allowed them to test. There was no multiple testing correction though they appeared to perform more than one test. There was no replication study to validate their original finding even if an independent replication is deemed required for the genetic association now.

Reviewer 2 Report

The authors investigated correlation between 11q11 copy number variation and obesity. It is an interesting topic. However, there are problems of experiment design and data analysis in this paper. 

There are 127 participants in this study. How were these 127 participants selected. According to my calculation, there might be some bias during the participant selection. For EA, there are 41 boys and 27 girls, while there are 28 boys and 31 girls for AA. A major finding of the paper is "High Olfactory Receptor-Rich 11q11 Copy Number in Girls and African American Children". If copy number is enriched in girls, the enrichment of copy number in AA might because there are more girls in AA than in EA, or vice versa. To solve this problem, the author should compare girls to boys within each ethnic group and compare EA to AA for boys and girls separately. However, the significance can only be found in Figure 1A and 2A instead of 3A. The major finding of the paper cannot be supported by the results. 

In genetic analysis, we usually do not combine different ethnic groups together or we adjust ethnic factors if we know the ethnic information to avoid population stratification bias. Please check table 1 of the paper https://www.ncbi.nlm.nih.gov/pmc/articles/PMC6007879/ for details. The authors should stratify the analysis.

The authors divided the participants into two groups, LCNV and HCNV, according to the median copy number (1.93). It is not a good idea. If the measurement of copy number of two samples are 1.9 and 2.1, both two samples should have two copies of DNA. However, they would be separated into two different groups. A better way is dividing the samples into deletion group and normal or duplication group. As shown in figure 1B, 0 and 1 are in a group and >=2 are in the other group. It is more biological meaningful. Nearest integer might not be a good idea. The authors may need to consult a Biostatistician or Bioinformatician. According to figure 1B, it is more reasonable to say CNV deletion is enriched in boys instead of CNV enriched in girls. It is normal to have two copies of DNA. 

There are errors in statistical analysis. In figure 1E, when the author analyze the data within LCNV group, all the calculation should be within the LCNV group. The percentage of NV girl + the percentage of OW/OB girl is less than 1(the percentage of NV boy + the percentage of OW/OB boy is more than 1), which indicates that the percentage is calculated between LCNV and HCNV instead of within LCNV group. The summation should be 1 if it is calculated within LCNV group. We will get unreasonable results if we apply the methods used by the authors to chromosome X. The percentage of NW and OW/OB for girls in LCNV would be 0. We will get the conclusion that the girls have higher chromosomes X CNV due to increased HCNV in OW/OB participants.  Same issues for figure 2 and 3. I am also confused there are two p values in figure 1D. It should a simple Fisher's exact test. There should be only one p value. The authors should ask a statistician for help. 

Minors

If p value is very small, for example 1e-5, we can write it as p < 0.001. For some large p values, for example p = 0.22, we write it as p = 0.22 in stead of p < 0.22. 

Line 136, abbreviation of "NW" was first used here. Please spell out the full term and put the abbreviation in parentheses.